# Plant-Based Diet, Cholesterol, and Risk of Gallstone Disease: A Prospective Study

**DOI:** 10.3390/nu11020335

**Published:** 2019-02-04

**Authors:** Chun-Ming Chang, Tina H. T. Chiu, Chia-Chen Chang, Ming-Nan Lin, Chin-Lon Lin

**Affiliations:** 1Department of General Surgery, Buddhist Tzu Chi General Hospital, Buddhist Tzu Chi Medical Foundation, Hualien 970, Taiwan; ccmjim1008@gmail.com; 2College of Medicine, Tzu Chi University, Hualien 970, Taiwan; 3Department of Nutritional Science, Fu-Jen Catholic University, Taipei 242, Taiwan; tina925@gmail.com; 4Department of Nutrition Therapy, Dalin Tzu Chi Hospital, Buddhist Tzu Chi Medical Foundation, Chiayi 622, Taiwan; 5Department of Medical Research, Dalin Tzu Chi Hospital, Buddhist Tzu Chi Medical Foundation, Chiayi 622, Taiwan; justenjoy555@gmail.com; 6Department of Family Medicine, Dalin Tzu Chi Hospital, Buddhist Tzu Chi Medical Foundation, Chiayi 622, Taiwan; mingnan.lin@gmail.com; 7Department of Family Medicine, College of Medicine, Tzu Chi University, Hualien 970, Taiwan; 8Department of Cardiology, Dalin Tzu Chi Hospital, Buddhist Tzu Chi Medical Foundation, Chiayi 622, Taiwan; 9Department of Internal Medicine, College of Medicine, Tzu Chi University, Hualien 970, Taiwan

**Keywords:** vegetarian diet, gallstone disease, cholesterol, prospective cohort

## Abstract

Vegetarian diets may lower symptomatic gallstone disease via cholesterol lowering. This study aimed to examine the risk of symptomatic gallstone disease (GSD) in Taiwanese vegetarians vs. nonvegetarians in a prospective cohort and to explore if this association is related to cholesterol concentration. We prospectively followed 4839 participants, and in the 29,295 person-years of follow-up, 104 new incident GSD cases were confirmed. Diet was assessed through a validated food frequency questionnaire. Symptomatic GSD was ascertained through linkage to the Taiwan National Health Insurance Research Database. Blood cholesterol profiles were measured at recruitment. Cox regression was applied to assess the effect of diet on symptomatic GSD, adjusting for age, education, smoking, alcohol, physical activities, diabetes, kidney diseases, body mass index, lipid-lowering medication, and hypercholesterolemia. Vegetarian diet was associated with a decreased risk of symptomatic GSD compared with nonvegetarian diet in women (hazard ratio [HR], 0.52; 95% confidence interval [CI], 0.28–0.96) but not in men. In women, nonvegetarians with hypercholesterolemia had 3.8 times the risk of GSD compared with vegetarians with normal cholesterol (HR, 3.81, 95% CI, 1.61–9.01). A vegetarian diet may therefore protect against GSD independent of baseline hypercholesterolemia. A nonvegetarian diet and hypercholesterolemia may have an additive effect in increasing GSD risk in women.

## 1. Introduction

Gallstone disease (GSD) is a worldwide disease. Its prevalence appears to be higher in Western countries (>10%) [1,2] than in Asian countries (3%–10%) [1,3]. Female sex, older age, higher body mass index (BMI), hyperlipidemia, alcohol consumption, and diabetes mellitus have been reported as risk factors for GSD [1]. In Western countries, gallstones are composed mainly of cholesterol in 75%–80% of cases [4].

The associations between GSD and plant-based foods have been explored in previous literature; however, the results were controversial. Some studies have suggested that fruit and vegetable consumption decreases the risk of cholecystectomy in women [5,6,7,8], but the European EPIC-Oxford study showed that a vegetarian diet is a risk factor for symptomatic GSD [9]. Most of these studies are from the Western population, so the effect of vegetarian dietary patterns on the development of symptomatic GSD in the Asian population is unknown.

The pathologic factors for cholesterol gallstones include genetic background, hepatic hypersecretion of cholesterol, and supersaturated bile, which give life to precipitating cholesterol crystals that accumulate and grow in a sluggish gallbladder [4]. The relationship between blood cholesterol levels and gallstone formation is multifactorial and complex. The role of blood cholesterol levels in the development of symptomatic GSD is not yet well known. 

Systemic review and meta-analysis studies have provided evidence that vegetarian diets effectively lower blood concentrations of total cholesterol (TCH), low-density lipoprotein (LDL) cholesterol, and high-density lipoprotein (HDL) cholesterol [10,11,12]. Vegetarian dietary patterns have been associated with reductions in risk of several diseases, such as hypertension, metabolic syndrome, diabetes mellitus, ischemic heart disease, diverticular disease, and colorectal cancer [13,14,15,16,17]. A vegetarian diet might be a useful nonpharmaceutical means for managing dyslipidemia, especially hypercholesterolemia [10].

We hypothesized that a vegetarian diet lowers the incidence of symptomatic GSD due to lowering of blood cholesterol. The purpose of this study was to examine the effect of vegetarian vs. nonvegetarian diet on the incidence of symptomatic GSD using a prospective cohort study in Taiwan and to explore if this association varies by cholesterol concentration.

## 2. Materials and Methods

### 2.1. Study Design and Population

The Tzu Chi Health Study is a prospective cohort study that recruited 6002 participants (age 18 to 87) from 2007 to 2009 at the Buddhist Dalin Tzu Chi Hospital. About 77% of the study participants were Tzu Chi volunteers. Tzu Chi volunteers are devoted Buddhist who volunteer regularly for the Buddhist Tzu Chi Foundation for a variety of charity works and disaster relief. These volunteers are encouraged, though not required, to become vegetarians.

### 2.2. Assessment of Diet, Cholesterol, and Other Covariates

At the time of enrollment, all participants received a comprehensive health examination and were interviewed on a questionnaire for demographics (sex, education), medical history, lifestyle habits (smoking, alcohol, physical activities), and diet (a questionnaire on vegetarian dietary practice [type and duration] and a food frequency questionnaire [FFQ]). 

Participants were interviewed on a 64-item quantitative FFQ that had been validated to show good reliability and validity in the present study population [18]. Vegetarian diet was defined as avoidance of meat and fish as assessed by both face-to-face confirmation and the FFQ by trained interviewers. 

Blood was drawn for health examination after an overnight fast. Total cholesterol, LDL cholesterol, and HDL cholesterol were assessed using Integra 800 System (Roche Diagnostics, Indianapolis, IN). Participants were categorized into two cholesterol levels: hypercholesterolemia (≧200 mg/dL) or normal cholesterol (<200 mg/dL). Height and weight were measured using an electronic scale and were used to calculate the BMI. BMI levels were further subdivided to two categories by 24 kg/m^2^, according to Taiwan’s definition of overweight and obesity [19]. 

### 2.3. Cohort Follow-Up and Case Ascertainment

The baseline data of the participants were linked to the National Health Insurance Research Database (NHIRD) of Taiwan. Incident cases of gallstone disease were identified through the International Classification of Diseases, Ninth Revision (ICD-9) codes 574, 575.0, and 575.1 in the NHIRD. The NHIRD covers medical benefit claims for over 23 million people (approximately 99% of Taiwan’s population) [20]. Taiwan’s National Health Insurance (NHI) provides universal insurance coverage and is a single-payer system with the government as the sole insurer. The database was monitored for completeness and accuracy by Taiwan’s Department of Health. 

Participants with any cancer or acquired immunodeficiency syndrome (*n* = 932), those who did not have a complete dietary and lifestyle questionnaire or were aged below 20 at recruitment (*n* = 30), and those who already had prior gallstone disease as identified by ICD-9 code in NHIRD before recruitment (*n* = 201) were excluded (Figure 1). 

Participants were followed from recruitment until December 31, 2014. During this period, gallstone diseases that were coded at least three times in the outpatient records or at least one time for inpatient treatment were considered as incident cases; this was done because the first couple of diagnoses in outpatient clinics may be made to enable further examination for rule out purposes and therefore may not be the true diagnosis. Person-years were calculated from the date of recruitment until the date GSD was coded for the first time in NHIRD or until the end of follow-up (death or December 31, 2014).

We identified participants’ comorbidities through the NHIRD using ICD-9 codes: hypertension (ICD-9 codes 401, 402, 405), heart disease (ICD-9 codes 410-414), diabetes (ICD-9 code 250), renal diseases (ICD-9 codes 403, 404, 580-588), and liver disease (ICD-9 codes 571, 572). Prescription of lipid-lowering medications such as statins was also determined through NHIRD. 

### 2.4. Ethics Statements

The study was approved by the Institutional Review Board for ethics in the Buddhist Dalin Tzu Chi Hospital. All participants gave written informed consents. The study has been registered at ClinicalTrials.gov (ID: NCT: 03204552).

### 2.5. Statistical Analysis

The SAS 9.4 (SAS Institute, Cary, NC, USA) was used for data analysis. A *p*-value of less than 0.05 was considered statistically significant. Because sex is an important predictor of GSD, we stratified all of our analyses by sex.

Baseline characteristics between vegetarians and nonvegetarians were compared using independent sample *t*-test (continuous variables) and chi-square test (categorical variables). Dietary intakes of vegetarians and nonvegetarians were compared using Wilcoxon sign rank test. 

Cox proportional hazards regression was used to calculate the risk of GSD with adjustments for age, education, drinking, smoking, diabetes and kidney diseases, lipid-lowering medications, TCH and BMI. We also conducted a subgroup analysis by cholesterol concentrations. 

A sensitivity test was conducted by eliminating the cases of symptomatic GSD diagnosed within one year of enrollment.

## 3. Results

### 3.1. Baseline Characteristics

A total of 4839 participants were included in the current study. Among them, 1403 were vegetarians, and 3436 were nonvegetarians. Their characteristics are summarized in Table 1. Vegetarians were slightly older, with a higher percentage of female, never-smokers, and habitual alcohol drinkers. Vegetarians had a lower prevalence of diabetes and kidney disease. Vegetarians also had lower cholesterol concentrations and BMI. 

### 3.2. Dietary Intakes of Vegetarians and Nonvegetarians

Dietary intakes of vegetarians and nonvegetarians, as assessed by FFQ, are shown in Table 2. While completely avoiding meat and fish, vegetarians had higher intakes of soy and vegetables. Intakes of dairy (median = 0.2 serving/day) and eggs (median = 0.3 serving/day) were low for both nonvegetarians and vegetarians. Vegetarians consumed more carbohydrates and fiber but less fat and protein compared with nonvegetarians. 

### 3.3. Vegetarian Diet and Gallstone Disease

In the 29,295 person-years (average 6.05 years) of follow-up, 104 individuals developed GSD. Among them, 22 were vegetarians, and 82 were nonvegetarians. Symptomatic GSD occurred at a higher incidence in nonvegetarians than in vegetarians (2.3% versus 1.5%).

Table 3 shows the association between a vegetarian diet and symptomatic GSD in Cox proportional hazards regression analysis. After adjustment for age, education, smoking, alcohol drinking, physical activities, diabetes, kidney diseases, BMI, hypercholesterolemia, lipid-lowering medication, and menopause (in women only), vegetarian diet was found to be associated with a nearly 50% decrease in symptomatic GSD risk compared with nonvegetarian diet (hazard ratio [HR], 0.52; 95% confidence interval [CI], 0.28–0.96) in women but not in men. In the whole population, age and hypercholesterolemia were two risk factors for symptomatic GSD (HR, 1.03; 95% CI, 1.01–1.06; and HR, 1.69; 95% CI, 1.12–2.55, respectively). In men, age (HR, 1.04; 95% CI, 1.01–1.08), hypercholesterolemia (HR, 1.89; 95% CI, 1.00–3.59), and diabetes (HR, 2.25; 95% CI, 1.01–5.03) were significantly associated with GSD.

In our subgroup analysis by cholesterol concentration (Table 4), we found vegetarian women had a 66% decrease in symptomatic GSD risk compared with nonvegetarian women (HR, 0.34; 95% CI, 0.14–0.82) among those with normal TCH but only a nonsignificant 23% (HR, 0.77; 95% CI, 0.33–1.79) among those with hypercholesterolemia, though the interaction of cholesterol concentration and vegetarian diet was not statistically significant (*p*-interaction = 0.365). 

The combined effect of a nonvegetarian diet and hypercholesterolemia is shown in Table 5. After adjusting for age, education, smoking, alcohol, physical activities, diabetes, kidney diseases, BMI, and prescription of lipid-lowering medications, we found that nonvegetarian women with TCH ≧ 200 mg/dL had 3.8 times the risk of developing symptomatic GSD compared with vegetarian women with a normal TCH (HR, 3.81; 95% CI, 1.61–9.01).

Sensitivity analysis of eliminating GSD diagnosed within one year of recruitment showed a similar protective association between vegetarian diet and GSD (HR, 0.42; 95% CI, 0.14–0.90) (Appendix A).

## 4. Discussion

In this prospective cohort study, we found that a vegetarian diet has an inverse association with symptomatic GSD in women but not in men. This protective association in women persists after adjustment for overweight/obesity and hypercholesterolemia. The potential protective effect of a vegetarian diet is most significant among those with normal cholesterol concentration. Nonvegetarian diet and hypercholesterolemia may have an additive effect in promoting GSD development. 

### 4.1. Vegetarian Diet and Gallstone Disease as Related to Cholesterol and Metabolic Risk Factors

One potential reason that a vegetarian diet may contribute to a lower GSD risk may be through lowering of cholesterol. Both vegetarian diets and plant protein have been shown to reduce cholesterol levels in meta-analyses [10,12,21]. In our study, we noticed that high total cholesterol concentration (TCH > 200 mg/dL) is a risk factor for symptomatic gallstone disease. Cholesterol, lecithin, and bile acid are the major components in bile [22]. Cholesterol supersaturation of the gallbladder bile is a necessary cause for gallstone formation [23]. Unphysiological supersaturation, generally from hypersecretion of cholesterol, is essential for the formation of cholesterol gallstones [24]. Previous studies have found a positive correlation between blood TCH, bile cholesterol [25], and cholesterol stone formation [26]. Although some studies have shown that abnormal serum lipid profile does not seem to be an essential feature for cholesterol GSD, patients with cholesterol GSD are more likely to have serum lipid parameters toward the undesirable cutoff levels of their respective normal ranges [27,28]. Our study found that vegetarian female with TCH < 200 mg/dL was a strong protective factor for symptomatic GSD. This suggests that even among those with normal TCH concentrations, adopting a healthy vegetarian diet could exert additional protection toward GSD. On the other hand, high cholesterol may attenuate the health benefit typically expected of a vegetarian diet. 

Besides cholesterol, GSD is also associated with other metabolic risk factors, including physical inactivity, insulin resistance, diabetes mellitus, and obesity [4]. Vegetarian diets have been reported to reduce insulin resistance [29] and body weight [30] and lower the risk of diabetes [16,31]. Nevertheless, we found that even after adjustment for cholesterol, BMI, diabetes, and physical activities, vegetarian diet was still associated with a lower risk of GSD in women. This suggests that a vegetarian diet may exert protection beyond cholesterol and these metabolic risk factors among women.

### 4.2. Vegetarian Diet and Gallstone Disease as Related to Female Sex 

Epidemiological and clinical studies have found that GSD is more common in women than men [2,32]. Studies have found that higher estrogen level increases the risk of GSD [4,32,33,34,35]. It has been proposed that estrogen increases the risk of developing cholesterol gallstones by increasing the hepatic secretion of biliary cholesterol, which leads to an increase in cholesterol saturation of bile [32]. Previous studies have suggested that in premenopausal women, low-fat diets [36,37,38] and low meat diets [39,40] may reduce serum estrogens levels. In postmenopausal women, vegetarians have been shown to have higher concentrations of sex hormone-binding globulin (SHBG, which inhibits sex hormones and reduces their bioavailability) and a lower level of free estradiol than omnivores [41,42]. This may explain, in part, the lower risk of developing symptomatic GSD in vegetarian females in our study. Our study also demonstrated that menopause was a protective factor for GSD, and this finding is consistent with the understanding that estrogen is a risk factor for GSD.

In contrast, the EPIC-Oxford study found a positive association between a vegetarian diet and GSD after BMI adjustment [9]. In their study, the vegetarian population was much younger than the nonvegetarian group: 42 vs. 50 years of age in men and 38 vs. 46 years of age in women. The significant age gap may have potentially induced bias through the difference in estrogen levels in females as menopausal status was not adjusted.

It has been reported that GSD patients have delayed intestine transit time and higher small intestine bacterial overgrowth [43]. Studies have shown that women’s intestinal transits are slower than men’s [44,45]. A slow colonic transit may, in turn, increase the risk of GSD by increasing the intestinal absorption of cholesterol [46]. Vegetarian diets are rich in dietary fiber, which may shorten the intestinal transit time [47] and may partly contribute to a lower risk of GSD.

### 4.3. Strength and Limitation

The strength of this study is the availability of participants’ biomarkers at recruitment. This enabled us to examine the associations with potential mediators, such as blood cholesterol concentration. Moreover, the diagnosis of GSD in this cohort study was based on NHIRD records, which is based on physician’s clinical diagnosis rather than self-reported medical history or sonography result. Sonography reports may indicate the existence of gallstone or sludge but are less clinically significant; it does not refer to symptomatic GSD as up to 80% of gallstone cases may remain asymptomatic [48,49,50]. 

There are some limitations to this study. First, the diagnosis of patients’ symptomatic GSD and identification of comorbidities were completely dependent on ICD codes, although the NHI Bureau of Taiwan randomly reviews the charts and interviews patients in order to verify diagnostic accuracy. In addition, the use of three outpatient diagnosis records for case ascertainment (stringent criteria) to avoid uncertainty and misdiagnosis such as biliary colic or acute gastritis in order to decrease false positive rate may have led to an under-identification of cases. Second, the follow-up period was only from five to seven years for the participants; nevertheless, the significant association suggests adequate power. Third, change in diet over time is possible, but only baseline diet pattern was available in the current study. However, any misclassification due to change in diet over time would tend to attenuate the results toward the null hypothesis. Fourth, the length of time as a vegetarian before recruitment was not certain. Therefore, we did the sensitivity test by eliminating GSD diagnosed within one year of recruitment, and the result was robust. Finally, the study was conducted in Taiwanese Buddhists who generally consume a healthy diet (adequate amount of fiber and vegetables), with little to moderate amount of animal products compared with Western populations, so the generalizability to other population requires additional studies. However, the small amount of meat consumption in our nonvegetarians would have narrowed the dietary range, making it more difficult to detect any potential diet–disease association. Therefore, the actual effect of a vegetarian diet may be even stronger if compared with high meat diets typical of Western populations.

## 5. Conclusions

This prospective cohort study shows that a vegetarian diet is associated with a lower incidence of symptomatic GSD in Taiwanese women but not in men. A vegetarian diet may strongly protect against GSD in women with normal cholesterol concentration.

## Figures and Tables

**Figure 1 nutrients-11-00335-f001:**
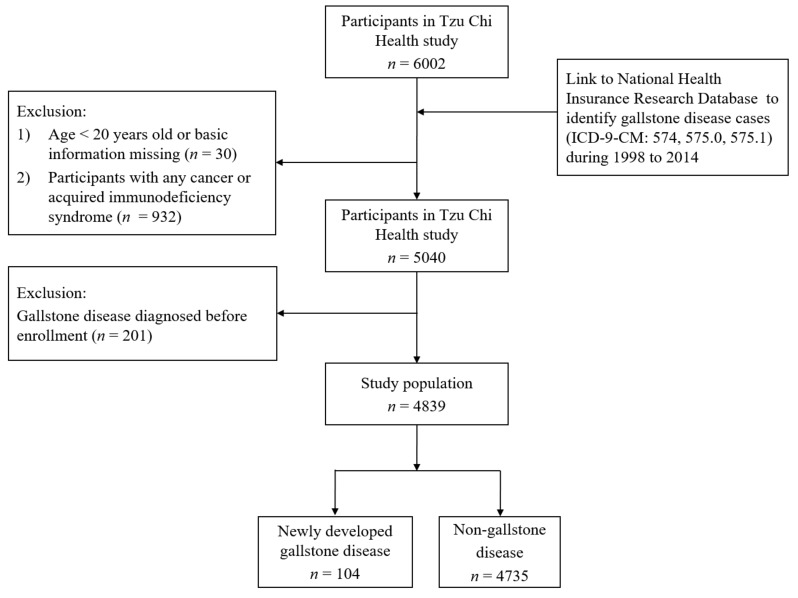
Flow chart for identifying the study cohort.

**Table 1 nutrients-11-00335-t001:** Baseline characteristics of vegetarians and nonvegetarians.

	All			Male			Female		
	Nonvegetarians	Vegetarians	*p*-Value	Nonvegetarians	Vegetarians	*p*-Value	Nonvegetarians	Vegetarians	*p*-Value
	*n* (%)/mean ± SD	*n* (%)/mean ± SD		*n* (%)/mean ± SD	*n* (%)/mean ± SD		*n* (%)/mean ± SD	*n* (%)/mean ± SD	
Age, mean ± SD years	51.8 ± 10.5	54.0 ± 9.3	<0.001	51.8 ± 10.9	54.2 ± 8.8	<0.001	51.7 ± 10.1	54.0 ± 9.4	<0.001
Sex			<0.001						
Male	1625 (47.3%)	345 (24.6%)							
Female	1811 (52.7%)	1058 (75.4%)							
Education degree			<0.001			<0.001			<0.001
Elementary school or less	733 (21.3%)	397 (28.3%)		232 (14.3%)	60 (17.4%)		501 (27.7%)	337 (31.9%)	
High school	1654 (48.2%)	719 (51.2%)		765 (47.1%)	178 (51.6%)		889 (49.1%)	541 (51.1%)	
College or higher	1049 (30.5%)	287 (20.5%)		628 (38.6%)	107 (31.0%)		421 (23.2%)	180 (17.0%)	
Smoking status			<0.001			<0.001			0.014
Ever	718 (20.9%)	116 (8.3%)		676 (41.6%)	105 (30.4%)		42 (2.3%)	11 (1.0%)	
Never	2718 (79.1%)	1287 (91.7%)		949 (58.4%)	240 (69.6%)		1769 (97.7%)	1047 (99.0%)	
Alcohol drinking status			<0.001			0.076			<0.001
Ever	614 (17.9%)	115 (8.2%)		539 (33.2%)	97 (28.1%)		75 (4.1%)	18 (1.7%)	
Never	2822 (82.1%)	1288 (91.8%)		1086 (66.8%)	248 (71.9%)		1736 (95.9%)	1040 (98.3%)	
Sport habit			0.003			0.428			0.098
Yes	2365 (68.8%)	904 (64.4%)		1176 (72.4%)	242 (70.1%)		1189 (65.7%)	662 (62.6%)	
No	1071 (31.2%)	499 (35.6%)		449 (27.6%)	103 (29.9%)		622 (34.3%)	396 (37.4%)	
Hypertension *			0.564			0.2641			0.471
Yes	752 (21.9%)	296 (21.1%)		391 (24.1%)	73 (21.2%)		361 (19.9%)	223 (21.1%)	
No	2684 (78.1%)	1107 (78.9%)		1234 (75.9%)	272 (78.8%)		1450 (80.1%)	835 (78.9%)	
Diabetes mellitus *			0.005			0.423			0.021
Yes	318 (9.3%)	95 (6.8%)		161 (9.9%)	29 (8.4%)		157 (8.7%)	66 (6.2%)	
No	3118 (90.7%)	1308 (93.2%)		1464 (90.1%)	316 (91.6%)		1654 (91.3%)	992 (93.8%)	
Heart disease *			0.461			0.186			0.642
Yes	338 (9.8%)	148 (10.5%)		172 (10.6%)	45 (13.0%)		166 (9.2%)	103 (9.7%)	
No	3098 (90.2%)	1255 (89.5%)		1453 (89.4%)	300 (87.0%)		1645 (90.8%)	955 (90.3%)	
Liver disease *			0.403			0.677			0.409
Yes	689 (20.1%)	266 (19.0%)		391 (24.1%)	79 (22.9%)		298 (16.5%)	187 (17.7%)	
No	2747 (79.9%)	1137 (81.0%)		1234 (75.9%)	266 (77.1%)		1513 (83.5%)	871 (82.3%)	
Kidney disease *			0.040			0.241			0.275
Yes	170 (4.9%)	50 (3.6%)		93 (5.7%)	14 (4.1%)		77 (4.3%)	36 (3.4%)	
No	3266 (95.1%)	1353 (96.4%)		1532 (94.3%)	331 (95.9%)		1734 (95.7%)	1022 (96.6%)	
Body mass index			<0.001			<0.001			<0.001
≧24	1611 (46.9%)	471 (33.6%)		914 (56.2%)	135 (39.1%)		697 (38.5%)	336 (31.8%)	
<24	1825 (53.1%)	932 (66.4%)		711 (43.8%)	210 (60.9%)		1114 (61.5%)	722 (68.2%)	
Total cholesterol			<0.001			<0.001			<0.001
≧200	1526 (44.4%)	369 (26.3%)		696 (42.8%)	69 (20.0%)		830 (45.8%)	300 (28.4%)	
<200	1910 (55.6%)	1034 (73.7%)		929(57.2%)	276 (80.0%)		981 (54.2%)	758 (71.6%)	
Lipid-lowering medications			<0.001			<0.001			<0.001
Yes	872 (25.4%)	224 (16.0%)		424 (26.1%)	56 (16.2%)		448 (24.7%)	168 (15.9%)	
No	2564 (74.6%)	1179 (84.0%)		1201 (73.9%)	289 (83.8%)		1363 (75.3%)	890 (84.1%)	
Menopause									<0.001
Yes							1032 (57.0%)	694 (65.6%)	
No							779 (43.0%)	364 (34.4%)	
Total cholesterol, mean ± SD	196.7 ± 36.7	180.7 ± 33.1	<0.001	194.0 ± 36.6	175.0 ± 34.4	<0.001	199.0 ± 36.6	183.0 ± 32.4	<0.001
HDL cholesterol, mean ± SD	54.2 ± 14.7	52.4 ± 13.7	<0.001	48.7 ± 12.4	44.9 ± 10.4	<0.001	59.1 ± 14.9	54.9 ± 13.7	<0.001
LDL cholesterol, mean ± SD	129.0 ± 33.4	116.0 ± 29.3	<0.001	130.0 ± 33.3	115.0 ± 28.1	<0.001	129.0 ± 33.5	117.0 ± 29.7	<0.001
Triglyceride, mean ± SD	117.0 ± 85.4	114.0 ± 77.8	0.230	133.0 ± 104.0	129.0 ± 93.6	0.481	102.0 ± 61.4	109.0 ± 71.3	0.014

* Comorbidity was diagnosed through the NHIRD using ICD-9 codes. Abbreviations: SD, standard deviation; NHIRD, National Health Insurance Research Database; ICD-9, International Classification of Diseases, Ninth Revision; HDL, high-density lipoprotein; LDL, low-density lipoprotein.

**Table 2 nutrients-11-00335-t002:** Dietary intakes of vegetarians and nonvegetarians as assessed by food frequency questionnaire.

Food and Nutrients	All			Male			Female		
	Nonvegetarians	Vegetarians	*p*-Value	Nonvegetarians	Vegetarians	*p*-Value	Nonvegetarians	Vegetarians	*p*-Value
	Median (P25, P75)	Median (P25, P75)		Median (P25, P75)	Median (P25, P75)		Median (P25, P75)	Median (P25, P75)	
Energy, kcal	1739 (1312, 2290)	1702 (1273, 2262)	0.118	2066 (1616, 2669)	2234 (1653, 2799)	0.068	1486 (1119, 1931)	1608 (1218, 2028)	<0.0001
Meat, servings	0.6 (0.2, 1.6)	-	-	1.0 (0.3, 2.2)	-	-	0.4 (0.1, 1.1)	-	-
Fish, servings	0.5 (0.1, 1.1)	-	-	0.6 (0.2, 1.3)	-	-	0.3 (0.1, 0.9)	-	-
Soy, servings	0.9 (0.5, 1.7)	1.5 (0.8, 2.5)	<0.0001	1.0 (0.5, 1.7)	1.6 (1.0, 2.9)	<0.0001	0.9 (0.5, 1.6)	1.4 (0.8, 2.3)	<0.0001
Dairy, servings	0.2 (0, 0.7)	0.2 (0, 0.6)	<0.0001	0.2 (0, 0.7)	0.2 (0, 0.7)	0.626	0.2 (0, 0.7)	0.2 (0, 0.6)	<0.0001
Eggs, servings	0.3 (0.1, 0.6)	0.3 (0.1, 0.4)	<0.0001	0.4 (0.2, 0.6)	0.3 (0.1, 0.5)	<0.0001	0.3 (0.1, 0.5)	0.3 (0.1, 0.4)	<0.0001
Vegetables, servings	3.8 (2.3, 5.7)	4.5 (2.9, 6.8)	<0.0001	3.5 (2.2, 5.5)	4.9 (3.0, 7.3)	<0.0001	3.9 (2.4, 5.9)	4.4 (2.9, 6.8)	<0.0001
Fruits, servings	1.0 (0.5, 2.0)	1.0 (0.6, 2.0)	0.020	1.0 (0.5, 2.0)	1.0 (0.6, 2.0)	0.112	1.0 (0.5, 2.0)	1.0 (0.5, 2.0)	0.226
Vitamin C, mg	164 (112, 233)	172 (118, 249)	0.001	165 (112, 226)	184 (130, 262)	<0.0001	163 (112, 237)	168 (115, 243)	0.191
Dietary fiber, g	19 (14, 26)	23 (16, 31)	<0.0001	19 (15, 27)	25 (19, 35)	<0.0001	19 (14, 26)	21 (16, 29)	<0.0001
Protein (% energy)	13 (12, 15)	12 (11, 13)	<0.001	13 (11, 15)	12 (10, 13)	<0.001	13 (12, 15)	12 (11, 14)	<0.001
Fat (% energy)	27 (21, 33)	25 (20, 30)	<0.001	26 (20, 32)	22 (18, 29)	<0.001	28 (22, 34)	25 (20, 30)	<0.001
Carbohydrate (% energy)	60 (54, 67)	64 (59, 70)	<0.001	61 (54, 68)	66 (59, 71)	<0.001	59 (53, 66)	64 (58, 69)	<0.001

One serving of meat, fish, soy is defined as 7 g of protein, one serving of eggs is defined as 1 regular size egg, one serving of vegetables is defined as 100 g, one serving of fruit is defined as 15 g of carbohydrates. *p*-value is assessed using Wilcoxon sign rank test. Abbreviations: P25, lowest 25%; P75, highest 75%.

**Table 3 nutrients-11-00335-t003:** Cox proportional hazards regression for risk of gallstone disease.

Cases/Person-Years	All	Male	Female
	104/29295	4311964	61/17332
	HR (95% CI)	HR (95% CI)	HR (95% CI)
***Crude model***	0.66 (0.41, 1.05)	1.08 (0.50, 2.33)	0.51 (0.28, 0.92)
***Adjusted model***			
Vegetarians vs. nonvegetarians	0.70 (0.43, 1.14)	1.26 (0.57, 2.77)	0.52 (0.28, 0.96)
Age	1.03 (1.01, 1.06)	1.04 (1.01, 1.08)	1.06 (1.02, 1.10)
Sex (male vs. female)	0.86 (0.53, 1.40)	-	-
High school vs. elementary	1.11 (0.68, 1.82)	1.58 (0.62, 3.99)	0.83 (0.44, 1.56)
College or higher vs. elementary	1.07 (0.59, 1.96)	1.67 (0.62, 4.52)	0.71 (0.30, 1.68)
Alcohol drinker	1.37 (0.76, 2.49)	1.68 (0.87, 3.23)	0.39 (0.05, 2.92)
Smoking	0.90 (0.48, 1.70)	0.78 (0.4, 1.54)	2.94 (0.68, 12.6)
Sport habit	1.01 (0.66, 1.56)	1.44 (0.66, 3.16)	0.89 (0.52, 1.52)
Diabetes mellitus	1.75 (0.98, 3.12)	2.25 (1.01, 5.03)	1.38 (0.59, 3.23)
Kidney disease	1.27 (0.58, 2.77)	0.66 (0.16, 2.76)	1.99 (0.78, 5.09)
BMI ≧ 24 kg/m^2^	1.36 (0.92, 2.03)	1.49 (0.79, 2.81)	1.35 (0.80, 2.27)
Lipid-lowering medications	0.82 (0.51, 1.32)	0.90 (0.44, 1.85)	0.72 (0.37, 1.39)
TCH ≧ 200 mg/dL	1.69 (1.12, 2.55)	1.89 (1.00, 3.59)	1.74 (1.00, 3.03)
Menopause	-	-	0.34 (0.16, 0.73)

Abbreviations: BMI, body mass index; HR, hazard ratio; CI, confidence interval; TCH, total cholesterol.

**Table 4 nutrients-11-00335-t004:** Subgroup analysis for risk of gallstone disease by total cholesterol concentration.

		All	Male	Female
**Total cholesterol ≧ 200 mg/dL**	Cases/person-years	56/11,421	23/4638	33/6783
	HR (95% CI)	0.89 (0.44, 1.81)	1.39 (0.41, 4.76)	0.77 (0.33, 1.79)
**Total cholesterol < 200 mg/dL**	Cases/person-years	48/17,874	20/7326	28/10,549
	HR (95% CI)	0.56 (0.28, 1.11)	1.31 (0.47, 3.70)	0.34 (0.14, 0.82)

Model adjusted for age, sex, education level, smoking status, drinking status, sport, diabetes, chronic kidney disease, BMI, lipid-lowering medications, menopause (for female). Abbreviations: HR, hazard ratio; CI, confidence interval.

**Table 5 nutrients-11-00335-t005:** Hazard ratio of diet with normal and high total cholesterol concentration for gallstone disease.

	All	Male	Female
Vegetarian with TCH < 200 mg/dL	1	1	1
Nonvegetarian with TCH < 200 mg/dL	1.72 (0.88, 3.33)	0.89 (0.32, 2.46)	2.48 (1.05, 5.88)
Vegetarian with TCH **≧** 200 mg/dL	2.20 (0.94, 5.14)	2.12 (0.50, 8.99)	2.64 (0.91, 7.65)
Nonvegetarian with TCH **≧** 200 mg/dL	2.71 (1.40, 5.21)	1.65 (0.60, 4.50)	3.81 (1.61, 9.01)

Adjusted for age, sex, education, smoking status, drinking status, sport, diabetes, chronic kidney disease, BMI, lipid-lowering medications, menopause (for female). Abbreviation: TCH, total cholesterol.

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
