# Peer review of "Plant-Based Diet, Cholesterol, and Risk of Gallstone Disease: A Prospective Study"

_nutrients, 2019, doi:10.3390/nu11020335_

Round 1

Reviewer 1 Report

This article reports the incidence of gallstone disease in a cohort of Taiwanese Buddhists, finding that a vegetarian diet is associated with reduced incidence in women, but not in men. A combination of a nonvegetarian diet and hypercholesterolemia was associated with increased risk in an additive fashion.

The article addresses an important clinical issue with an appropriate scientific method. It presents interesting and clinically useful findings. It is generally well-written and thoughtfully presented.

The cohort is well characterized. A notable strength of the study is that case ascertainment was done through linkage to the Taiwan National Health Insurance Research Database. In addition, dietary assessment was done carefully through questionnaires and face-to-face interviews.

Potential weaknesses include the fact that lipid levels were measured only at recruitment, as is common with cohort studies, and the follow-up was rather brief (six years). Neither of these is a major weakness, and the findings are nonetheless important.

It would be helpful for the authors to add a sentence describing the rationale for not confirming a diagnosis of gallstone disease until it had been coded three or more times in an outpatient record or one or more times on an inpatient record. The investigators may wish to describe whether the rather stringent requirement regarding outpatient treatment may have led to an underidentification of cases.

I would suggest removing the comments about animal studies (lines 221, 242, and 258). They do not provide helpful insights. Animals are well-known to have different cholesterol metabolism, compared with humans, the literature on gallstones in animals is very modest and not particularly useful in relation to human disease. A focus on human research would be more appropriate.

Line 228. The sentence about junk food is not supported by anything in the manuscript. I would suggest deleting it.

The language is good, but review by a native English speaker would be helpful. 

Author Response

Response to Reviewer 1 Comments

Dear editor and reviewers

Thank you very much for your prompt and helpful comments to our manuscript. Below, please find our point-to-point response to the comments raised by the reviewers.

Thank you very much again,

Yours Sincerely,

Chin-Lon Lin, MD

Point 1: It would be helpful for the authors to add a sentence describing the rationale for not confirming a diagnosis of gallstone disease until it had been coded three or more times in an outpatient record or one or more times on an inpatient record. The investigators may wish to describe whether the rather stringent requirement regarding outpatient treatment may have led to an underidentification of cases.

Response 1: Thank you for the suggestion. We have added what you suggested in

-        line 121 – 123, indicating that the first couple of diagnoses in outpatient clinics may be made to enable further examination for rule out purposes rather than true diagnosis

-        line 363 – 365, indicating that the used of three outpatient diagnosis records for case ascertainment (a stringent criteria) to avoid uncertain and misdiagnosis such as biliary colic or acute gastritis in order to decrease false positive rate may have led to an under-identification of cases

Point 2: I would suggest removing the comments about animal studies (lines 221, 242, and 258). They do not provide helpful insights.

Response 2: We have removed those comments about animal studies.

Point 3: Line 228. The sentence about junk food is not supported by anything in the manuscript. I would suggest deleting it.

Response 3: We have deleted this.

Point 4: The language is good, but review by a native English speaker would be helpful.

Response 4: Thank you. Hopefully the revised version has most if not all the language problems corrected.

Reviewer 2 Report

This is a very interesting and important review. I don't have any substantive edits; however, I do wonder if it might be worth mentioning how much healthier a non-vegetarian diet is in Taiwan vs. Western countries. I was struck by how much more fiber and little dairy and egg both groups ate, compared with Western non-vegetarians and even vegetarians. Even the meat servings for the non-vegetarians was less than one would see in a Western country. What would be the relevance of this to your work, if any?

Lines 33-34 seems to have either a missing or extra parenthesis. In this sentence: Vegetarian diet was associated with decrease risk for 33 symptomatic GSD compared to non-vegetarian, hazard ratio (HR), 0.52; 95% confidence interval (CI), 34 0.28-0.96) in women but not men.

Line 41 say "Western" as opposed to "the Western"

Line 44 change to "composed mainly of" so remove one "of"

There are many grammatical errors that are not substantive, such as the two above, so the authors/editors may want to have this paper reviewed by a native speaker. I'm going to stop pointing them out in the interest of time.

Author Response

Response to Reviewer 2 Comments

Dear editor and reviewers

Thank you very much for your prompt and helpful comments to our manuscript. Below, please find our point-to-point response to the comments raised by the reviewers.

Thank you very much again,

Yours Sincerely,

Chin-Lon Lin, MD

Point 1: I do wonder if it might be worth mentioning how much healthier a non-vegetarian diet is in Taiwan vs. Western countries. I was struck by how much more fiber and little dairy and egg both groups ate, compared with Western non-vegetarians and even vegetarians. Even the meat servings for the non-vegetarians was less than one would see in a Western country. What would be the relevance of this to your work, if any?

Response 1: Thank you for the suggestion. One problem of comparing food intakes in different studies is that each study uses a different food frequency questionnaire (FFQ). The FFQ method is excellent for ranking dietary intakes (relative validity) but not assessing absolute intakes. Therefore, it is difficult to compare actual intakes without additional effort in harmonization. However, we agree with the reviewer’s comments and believe the meat intake in our population is much lower than the general population since most of nonvegetarians would consider themselves part time vegetarians. We think this would actually underestimate the results and have addressed this in line 371 – 377.

Point 2: There are many grammatical errors that are not substantive, such as the two above, so the authors/editors may want to have this paper reviewed by a native speaker.

Response 2: We have edited the English writing. Thank you very much.
